# Human Postures Recognition by Accelerometer Sensor and ML Architecture Integrated in Embedded Platforms: Benchmarking and Performance Evaluation

**DOI:** 10.3390/s23021039

**Published:** 2023-01-16

**Authors:** Alessandro Leone, Gabriele Rescio, Andrea Caroppo, Pietro Siciliano, Andrea Manni

**Affiliations:** National Research Council of Italy, Institute for Microelectronics and Microsystems, 73100 Lecce, Italy

**Keywords:** ageing adults, posture classification, wearable sensor, machine learning, embedded platform

## Abstract

Embedded hardware systems, such as wearable devices, are widely used for health status monitoring of ageing people to improve their well-being. In this context, it becomes increasingly important to develop portable, easy-to-use, compact, and energy-efficient hardware-software platforms, to enhance the level of usability and promote their deployment. With this purpose an automatic tri-axial accelerometer-based system for postural recognition has been developed, useful in detecting potential inappropriate behavioral habits for the elderly. Systems in the literature and on the market for this type of analysis mostly use personal computers with high computing resources, which are not easily portable and have high power consumption. To overcome these limitations, a real-time posture recognition Machine Learning algorithm was developed and optimized that could perform highly on platforms with low computational capacity and power consumption. The software was integrated and tested on two low-cost embedded platform (Raspberry Pi 4 and Odroid N2+). The experimentation stage was performed on various Machine Learning pre-trained classifiers using data of seven elderly users. The preliminary results showed an activity classification accuracy of about 98% for the four analyzed postures (Standing, Sitting, Bending, and Lying down), with similar accuracy and a computational load as the state-of-the-art classifiers running on personal computers.

## 1. Introduction

In recent years, especially in developed countries, the average age of the population is increasing more and more; therefore one of the most important challenge is to try to improve the quality of life of the elderly population. In fact, according to recent European Union (EU) statistics [1] elderly are expected to constitute an increasing share of the total population in the period between 2021 and 2100 and, in particular, people aged 65 years will increase from 20.8% in 2021 to 31.3% of the EU population by 2100. To address these predictions, intelligent systems are being developed to improve the lifestyle of elderly and enable them to live actively and independently, against the limitations of age. Consequently, smart health is gaining importance in recent years due to the rapid development of purpose-built sensor systems [2,3,4].

To understand a person’s lifestyle, it is important to examine their daily activities. Indeed, using such data, it is possible to understand people’s activity patterns and, consequently, implement measures to improve their lifestyle.

For activity monitoring, which is a specific research area of Ambient Assisted Living (AAL) systems for elderly people, posture can be used. Posture is based on the position of the skeleton and muscles during various daily activities [5]. In addition to monitoring the individuals’ activities, postural analysis is essential to recognize incorrect postural habits that are the cause of various problems, such as back pain, shoulder pain, scoliosis, etc. In these cases, people are required to resort to medication and physical treatment, with the resulting discomfort, which can be avoided if the problem is identified in a timely way [6,7]. This analysis can be performed in two ways: posture detection and posture recognition [8]. In the first case, a person’s posture is classified as good or bad on the basis of current activity. In contrast, in posture recognition, people’s activity patterns are analyzed to find out whether they have a healthy lifestyle or not.

Two major approaches in the literature for posture recognition and detection are used: computer vision or image processing techniques and wearable sensors based methods. The first technique uses a single camera [9] or stereo vision [10,11]. These methods, first of all, capture the principal areas of the body, such as the head, and the extremities of the body (hands and feet) and the secondary parts of the body, such as the shoulders and the joints (the elbows and knees). For example, fall detection based on 3D head movement trajectory was analysed in [12]. Specifically, the motion trajectory of the head is obtained through a layered particle filter with 4 layers and 250 particles, and through this trajectory the fall is detected. The system uses a single camera to complete the 3D calculation of the head. A vision-based fall detection method is proposed in [13]. In this approach, the histogram formed by the elliptic central coordinates of the human body is used to count the eigenvalues of the human body image. Then the eigenvalues are used to train a Support Vector Machine (SVM) algorithm for posture classification. An approach for posture classification is presented in [14], consisting in three phases. In the first phase, the human silhouette is extracted from the input video frames. The second stage extracts local and global features from the human silhouette. Finally, these features are used for posture classification, discriminating between normal and abnormal posture. However the approaches based on computer vision techniques present several drawbacks. One of these is related to the positioning of the camera in each room in which the system must operate. Another problem is its operation only in indoor environments. Other common disadvantages are the possible occlusion of the target and the distance from the camera. In fact, as the distance between the camera and the person increases, the quality of the acquisition accuracy decreases. However, the main problem of this approach is related to the privacy of the user, since it is necessary that an observed subject agrees to be captured by a camera.

Compared to the camera-based approach, wearable sensors do not have the problem of violating end users’ privacy, an extremely important issue in health-related applications. Another advantage of using a posture recognition system based on wearable devices is that it can be used both indoors and outdoors via wireless communication technology. Wearable sensor-based methods are realised in different ways, e.g., the pressure distribution on different surfaces is calculated or the angular difference between the current posture and a predicted correct posture is computed. The first approach is used for example in [15], where force and ultrasound sensors are used. However, the limit of this approach is that it can only be used for a restricted kind of postures, such as sitting. In the second approach, wearable sensors such as accelerometers and gyroscopes are used to detect posture. Specifically, accelerometers measure acceleration along the x, y and z axes, while gyroscopes determine the angular velocity variation along the x, y and z axes. Such values are subsequently sent to a computing unit via wireless (WiFi or Bluetooth) for processing and classification. These sensors can be easily applied to various parts of the body. This second approach is much investigated in the literature. For example, an algorithm for improving posture using a triaxial accelerometer and a gyroscope via two sensors is presented in [16]. Various classifiers such as Decision Tree (DT), Random Forest (RF), SVM and perceptrons are compared to achieve better accuracy. In [17], a wearable posture identification system using two accelerometers located in two different areas of the human spinal column (human lumbar spine and human cervical spine) is presented to monitor and identify the good and bad sitting posture, while an Arduino system was considered to calculate the angle to determine the posture. In [18], authors classify human posture during three different activities (standing, sitting and sleeping/lying) as healthy or abnormal. A large dateset obtained through the use of accelerometric and gyroscopic data collected by MPU-6050 sensors mounted at different body positions is used. Machine Learning (ML) techniques are applied to this data. Ref. [19] employs inertial sensors for human posture recognition. After human posture data are collected, a hierarchical multilevel human posture recognition algorithm is used. The algorithm pre-processes the posture signal, removes the outliers, extracts and selects the posture signal with the posture identifier. In [20] a Deep Neural Network (DNN) consisting of a three-layer convolutional neural network followed by a long short-term memory layer was analyzed to classify six activity types for hospitalized patients. In order to collect data, a single triaxial accelerometer attached to the trunk is used. To compare DNN accuracy, a SVM algorithm is trained on collected accelerometer data.

In the literature human postures recognition systems mostly use Personal Computers (PC), which are not easily transportable and have high energy consumption. To overcome these limitations, increase the level of usability and promote health monitoring of elderly, it becomes more important to develop human posture monitoring systems on portable, easy-to-use, compact, and energy-efficient embedded platforms. In fact, considering the requirements of future commercial applications, the market demand for embedded platforms is great. Few works in the literature have employed embedded systems in this context. For example, [21] uses a Raspberry Pi for processing six classes of human postures with an artificial neural networks (ANNs). The proposed methodology successfully classifies human postures with an overall accuracy of approximately 97.6%. In this work two wearable devices positioned on the chest and on the right thigh were used. In [22] seven types of sitting postures with a pressure sensor array are processed on a Raspberry Pi using seven ML algorithms for comparation, showing that a five-layer Artificial Neural Network achieves the highest accuracy of about 97%.

This work presents the design and implementation of a real-time posture recognition ML algorithm performing highly on embedded platforms with low computational capacity. This software was integrated and tested in two low-cost embedded platforms (Raspberry Pi 4 and Odroid N2+). Using these platforms offers an excellent cost-benefit ratio versus a traditional PC, considering energy consumption, price, lightweight, size, portability, and reliability.

The main contributions of the proposed work are listed below:an algorithmic framework for the classification of postures by using only one commercial wearable sensor is designed and implemented;three different ML classification algorithms are compared to distinguish between posture;a performance comparison of the proposed algorithm between a PC and previously mentioned embedded platforms demonstrated real-time operation on such platforms in terms of processing time, power consumption, and computation flexibility.

The remainder of this paper is organized as follows. Section 2 reports an overview of the algorithmic framework for human posture recognition. Performance results of the algorithms, evaluated on the embedded systems and compared with a PC, are presented in Section 3. Finally, Section 4 shows both our conclusions and discussions on some ideas for future work.

## 2. Materials and Methods

The proposed platform is composed of two components: a wearable device for collecting accelerometer data and embedded platforms for processing these raw data and extracting human postures. The use of embedded platforms makes it possible to meet the typical requirements of AAL applications. In fact, these platforms are equipped with features that enable easy integration into various AAL environments by being easily portable, low-priced, lightweight and reliability. Figure 1 reports the algorithmic pipeline via a block diagram.

### 2.1. Wearable System

For the evaluation of user movements suitable for posture recognition [23], a inertial based system was used and placed on the chest by means of a band. Specifically, the Shimmer3 IMU sensor device [24], which integrates a triaxial accelerometer, magnetometer, pressure/temperature sensor, and triaxial gyroscope, was adopted. For the evaluation of the user’s change of posture only the tri-axial accelerometer was considered, it measures the acceleration referred as Earth’s gravity “g” force (9.81 m/s${}^{2}$) and it is DC coupled. Thus, it is possible to evaluate both accelerations under static and dynamic conditions along the three axes. The Shimmer device is well suited for long-term monitoring, as exhibits a low degree of invasiveness since it is lightweight, small in size, and it is equipped with a low-power wireless (Bluetooth) connection for data transmission. The battery life in streaming mode is about 8 h. Table 1 shows the main characteristics of the Shimmer3 IMU device taking into account only the accelerometer sensor.

In addition, Shimmer devices are equipped with open software libraries that allow for data acquisition and the development of custom applications. The system was worn as shown in Figure 2, using the included band that provides a comfortable fit and good mechanical stability.

The data are acquired with a full scale in the range of 2 g and a sampling frequency set at 50 Hz, which is sufficient to identify human postures. Then, they are sent to the computing unit (PC or embedded platforms) on which the processing software is located.

The following paragraphs will briefly describe the procedures implemented for each step.

#### 2.1.1. Pre-Processing and Calibration Phases

The first phase of the framework aims to reduce electrical/environmental noise and obtain data in a format suitable for further processing. To this aim, first, acceleration data on three axes (Ax, Ay and Az) are read from the device worn by the user during data collection and converted to gravitational units, to represent acceleration data in the range ±2 g. This makes it possible to extract the α angle of chest tilt and avoids having too different orders of magnitude during subsequent processing steps. Next, the noise is filtered out through the use of a low-pass filter of order 8 and cut-off frequency 10 Hz.

A calibration method was performed to verify that the device is worn correctly and to store the starting settings, after the device is placed, in order to handle the pre-processed data correctly. The check is performed by analyzing the acceleration values on the three coordinated axes when the user is in a standing position and in a static condition. In summary, the check involves verifying that the measured values on two acceleration axes are orthogonal to g, i.e., have a value close to zero, less than a predefined tolerance range. After performing the check, the acceleration values on the three axes thus obtained are stored and used in subsequent processing steps to derive the initial sensor positioning conditions.

#### 2.1.2. Feature Extraction and Selection Phases

The data processed as described in the previous section are used to obtain the most suitable information for the identification of human postures. To this purpose, the main features in the literature developed in applications for monitoring the human posture were identified and analyzed [25,26,27,28,29]. The focus was on those achieved in the time domain in order to reduce computational cost and execution time. In Table 2 all features considered are listed.

These characteristics were calculated for each acceleration axis within a sliding window of 350 ms, with an incremental window of 50 ms. To reduce the complexity of signal processing and improve system performance, the Lasso feature selection method, suitable for supervised systems, was applied [30]. Through this technique, the following features were chosen: mean absolute value, variance, dynamic acceleration change, static acceleration change, kurtosis and skewness.

#### 2.1.3. Classification

After the feature extraction and selection phase, various ML algorithms were trained on the acquired data for comparison. In particular, three algorithms showing the best performance are reported: RF, DT and K-Nearest Neighbours (KNN).

RF algorithm [31] generates a set of predictors from decision trees using the hyperparameters of each tree. To classify the input vector, a vector independent of the input vector is used and each tree votes for the largest number of classes. This algorithm randomises the model by increasing the number of trees. In a random subset of features, the best feature is identified.

In DT [32], a certain parameter is used to partition the data. As predictive model, a tree is used to traverse the branches of the tree, which represent the observations on a feature, to reach the leaves, which represent the target value of the feature and, consequently, the class labels.

KNN [33] is a widely used method in classification due to its high performance and ease of implementation. In particular, each sample is assigned to a category if most of the k samples close to the one considered belong to the same category. The value of k is normally no higher than 20 [34]. Choosing the optimal value of k is important because if k is too small, noise may be present, while if k is too large, samples belonging to other classes may be present in the neighborhood.

A grid search technique [35] was applied to obtain the optimal parameters for each ML model. These parameters are shown in Table 3.

### 2.2. Elaboration Units

The proposed framework implementation was based on a desktop for the validation phase and on two embedded architectures for comparison. The embedded architectures used are: Raspberry Pi 4 Model B and Odroid N2+. Figure 3 shows elaboration units employed. Wearable sensor is connected to these units via Bluetooth protocol and algorithms for the acquisition and processing of raw data are implemented on units. Following are the characteristics of each processing unit involved in this study.

The Raspberry Pi [36] has Broadcom BCM2711, quad-core Cortex-A72 (ARM v8), 64-bit 1.5 GHz processor, 8 GB of RAM, LAN, Bluetooth 5.0, Gigabit Ethernet, 2 USB 3.0 and 2 USB 2.0, 40 general-purpose input/output (GPIO) pins and a Micro SD card slot for loading operating system and data storage. Its operating system is Raspbian, a Debian-based Linux distribution.

Odroid N2+ board [37] has a quad-core Cortex-A73 processor to 2.4 Ghz, 4 GB of RAM. It has 1 RJ45 Ethernet, 4 USB 3.0, 1 Micro USB2.0, 1 HDMI 2.0 and, as Raspberry, a Micro SD card slot for operating system and data storage. The operating system is Ubuntu. The Odroid board has a fan that turns on periodically, causing an increase in power consumption compared to the Raspberry.

As desktop, a PC was used. In particular Lenovo ThinkCentre M70s Tiny [38] with Intel Core i5 at 2.5 GHz as processor and 8 GB of RAM was used. It has 1 RJ45 Ethernet, 4 USB 2.0 and 4 USB 3.0, Bluetooth 5.0, 1 HDMI 2.1, HD SSD of 256 GB and Windows 10 as operating system.

Table 4 shows the main characteristics and differences of these elaboration units.

## 3. Results and Discussion

To validate the proposed framework and to verify its functioning in real time, a series of experiments were carried out on the processing platforms previously described. The validation was conducted in “Smart Living Technologies Laboratory” located in the Institute of Microelectronics and Microsystems (IMM) in Lecce, Italy. Due to COVID-19 restrictions, it was only possible to validate the entire platform with 7 ageing subjects with an average age of 66.13 ± 7.10 years old.

Considering the implementation aspect, the algorithm was coded in Python without using any Deep Learning libraries such as TensorFlow or Keras, which would be resource-intensive on embedded devices such as the Raspberry-Pi. Moreover, this choice also contributes to the robustness of the system in terms of minimal delay in real-time posture classification and independence from the use of any specific package. In particular, the unofficial Python API for Shimmer Sensor devices “pyshimmer” [39] was used to acquire accelerometer data, while the following libraries were used for data processing and classification: “numpy”, “pandas”, “scipy”, “pickle”, “more-itertools”.

Firstly the posture classification performance were evaluated using Accuracy (Acc), Precision (Pr), Recall (Re), F1-score as metrics. Then the benchmarking of the proposed pipeline on the described embedded platforms was analyzed.

Classifiers’ performance metrics are defined by the following expressions:(1)$$\begin{array}{ccc} \mathrm{Acc} & = & \frac{\mathrm{TP}+\mathrm{TN}}{\mathrm{TP}+\mathrm{TN}+\mathrm{FP}+\mathrm{FN}} \end{array}$$(2)$$\begin{array}{ccc} \mathrm{Pr} & = & \frac{\mathrm{TP}}{\mathrm{TP}+\mathrm{FP}} \end{array}$$(3)Re=TPTP+FN(4)$$\begin{array}{ccc} F1-\mathrm{score} & = & \frac{2*\mathrm{TP}}{2*\mathrm{TN}+\mathrm{FP}+\mathrm{FN}} \end{array}$$
where TP (True Positive) indicates samples that are correctly predicts as positive, TN (True Negative) represents samples that are correctly predicts as negative, FP (False Positive) denotes that negative sample are incorrectly predicts as positive and, finally, FN (False Negative) indicates that positive sample are incorrectly predicts as negative.

The performance of each ML model was compared on the basis of test sets. To this aim, a 10-cross-validation [40] was applied. Through this procedure, the training set of each classifier is perturbed by randomising the original dataset. Thus, each classifier is trained using 80% of the data, while testing is performed on the remaining 10% and 10% is used to build a validation set. The procedure is repeated 10 times making sure that the same samples do not simultaneously occur in the training and test sets.

In Table 5 the performance of each ML model are reported. RF model showed the best performance in terms of Acc, Pr, Re and F1-score obtaining in numerical terms an accuracy of above 98.7% with a performance improvement of about 1–4% in terms of accuracy compared to the other two classifiers.

Subsequently, the average time to obtain a posture was evaluated using the previously described classifiers to assess the real-time operation of the algorithm on the three considered hardware platforms. Table 6 shows the obtained values of average time varying classifiers and platforms. As we can see, considering the sensor’s sampling period (20 ms), the classifier with the best performance in terms of accuracy, on the embedded boards (Raspeberry Pi 4 and Odroid N2+) is not able to guarantee the real-time operation of the proposed solution. On the contrary, DT, whose performance is in any case comparable to RF, is the one achieving the lowest posture elaboration times on all three platforms considered, guaranteeing correct real-time operation.

To enable real time operation also for RF classifier, its performance was verified by downsampling at 25 Hz instead of the used 50 Hz. Table 7 shows the values obtained for the four previously described metrics. A decrease in average accuracy of the order of 2.5% can be seen, while still guaranteeing an acceptable trade-off between accuracy and real-time operation.

Benchmarks have a significant impact on the usability of a model in the target application. In this paper, the efficiency of the proposed human posture recognition technique is evaluated by measuring CPU load, memory utilization and power consumption on each considered hardware platform. The Python library “psutil” was used to evaluate CPU load and memory utilization and they are reported in percentages. The power consumption of each embedded board was measured using a USB Power Meter Tester UM25C [41] manufactured by RuiDeng and shown in Figure 4 and monitored via Android and/or iOS apps [42,43]. This provided an indication of the power consumption during the performed tests.

CPU load, memory utilization rate and power consumption were monitored by running 10 tests for each experiment. At the end of each evaluation, we calculated the average CPU load, memory utilization and power consumption for that particular experiment. Then, 10 values of CPU load, memory utilization and power consumption were generated. Finally, the results were averaged, giving the average of the three benchmarking values.

Table 8 shows the comparison of CPU load and memory usage between the three analyzed platforms.

In terms of CPU load, the trend on all platforms shows, as expected, a similar load with values between 0.147% and 0.332%. The maximum CPU load values can be seen on all platforms for RF with values between 0.312% and 0.332%, while DT shows the lowest CPU load with a range between 0.147% and 0.265%.

Trends in memory usage show fairly similar behaviour on all three considered hardware platforms and do not seem to be influenced by the platform architecture, being between 0.144% and 0.279% of its capacity.

With regard to power consumption on both embedded boards used for testing, the results are illustrated in Figure 5. During the tests, various peripherals (monitor, keyboard and mouse) were connected to the embedded boards, increasing the power consumption accordingly.

Raspberry Pi 4 showed the best results, with low power consumption that is probably due to the absence of a cooling unit. In comparison, the Odroid N2+ board showed the highest power consumption, at least partly due to its active cooling unit. This relatively higher power consumption compared to Raspberry, however, did not affect the board’s performance by maintaining a low CPU load and RAM utilization, giving the board an acceptable compromise between computation time and power consumption.

Finally, to demonstrate the effectiveness of the proposed solution on embedded boards, the ratio between the cost of the three analyzed platforms and the performance used for the benchmark was considered. Figure 6 shows the obtained values at varying the three ML classifiers considering, respectively, CPU load (a) and RAM utilization (b). As we can see the embedded boards achieve a significantly lower ratio considering their cost of about four times lower than the PC.

## 4. Conclusions

This paper presented a framework for human posture recognition using a real-time accelerometer sensor.

The objectives of our proposed work were mainly twofold. First of all, the effectiveness of the proposed approach in classifying human postures into four categories (sitting, standing, lying down and bending) was verified. To this end, three ML classification algorithms were evaluated, obtaining an overall accuracy of approximately 98.7%. The system is able to classify postures in real time with the help of an accelerometer placed on the users’ chest.

The second objective was to evaluate the implementation compatibility of the proposed approach on ARM-based embedded boards (Raspberry Pi 4 and Odroid N2+) for the real-time processing and classification of signals acquired by the involved accelerometer sensor. For this purpose, a comprehensive benchmark was performed, including CPU load, memory occupancy and power consumption.

As expected, the overall performance of the used platforms is equivalent considering both PC and embedded platforms. Furthermore, the ability of embedded platforms to match the performance of high-end platforms certainly indicates a good potential for use in AAL environments given their low cost compared to normal PCs (about four times lower in the analysed case) and thus easier deployment. Raspberry Pi 4 outperformed the two evaluated embedded boards, with the lowest power consumption, CPU load and memory occupancy similar to high-end platforms. On the contrary, Odroid N2+ showed a higher power consumption than Raspberry Pi 4 probably due to its active cooling unit, but nevertheless this higher power consumption is still acceptable given the obtained low CPU load and memory occupancy.

In conclusion, the following strengths of the whole designed and implemented system can be highlighted: (1) the algorithmic pipeline makes it possible to classify human postures by a wearable device readily available on the market, low cost and easy to use; (2) employing such sensor allows to avoid problems of violation of users’ privacy, an extremely important issue in health-related applications, also allows the use in both indoor and outdoor environments thanks to Bluetooth communication technology; (3) embedded platforms enable easy deployment in AAL environments, given their small size and low cost compared to a standard desktop PC. In addition, their low power consumption allows for increased portability by equipping them with a portable battery.

However, the present study has an important limitation, in particular, the number of elderly subjects involved was not high due to the pandemic situation. Therefore, the achieved classification results should be confirmed with a larger dataset of users.

Future work will include the use of other embedded boards such as, for example, other versions of Odroid or UP board [44] with Intel processor. In addition, other body positions for sensor placement such as, for example, the thigh or back will be tested. Furthermore, the combination of two wearable devices could be considered to increase the classification accuracy of some postures that might be confused with one device such as, for example, standing/sitting and bending/lying down. Finally, a larger dataset of user will be considered to validate the classification results.

## Figures and Tables

**Figure 1 sensors-23-01039-f001:**
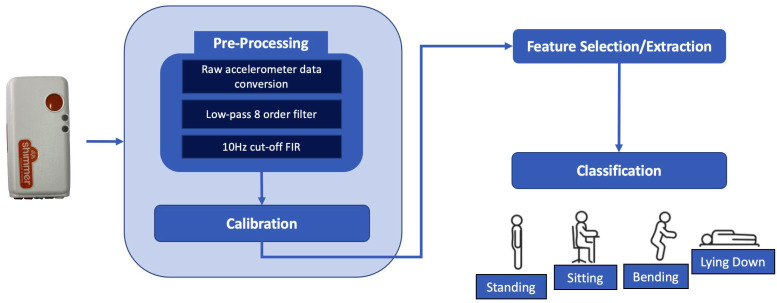
Main steps of software framework via a block diagram.

**Figure 2 sensors-23-01039-f002:**
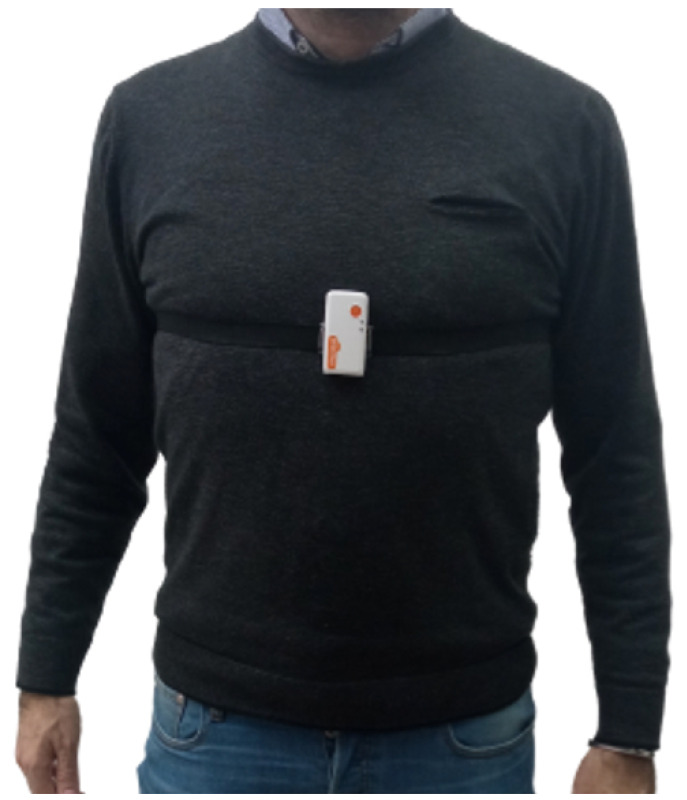
Sensor device positioning.

**Figure 3 sensors-23-01039-f003:**
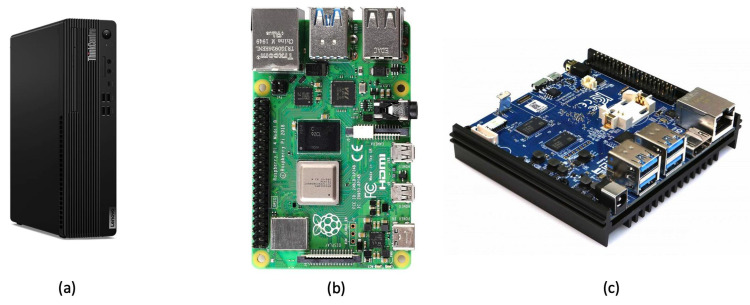
Elaboration units (Lenovo ThinkCentre (**a**), Raspberry Pi 4 Model B (**b**), Odroid N2+ (**c**)) for the acquisition, processing accelorometer data and classification of human postures.

**Figure 4 sensors-23-01039-f004:**
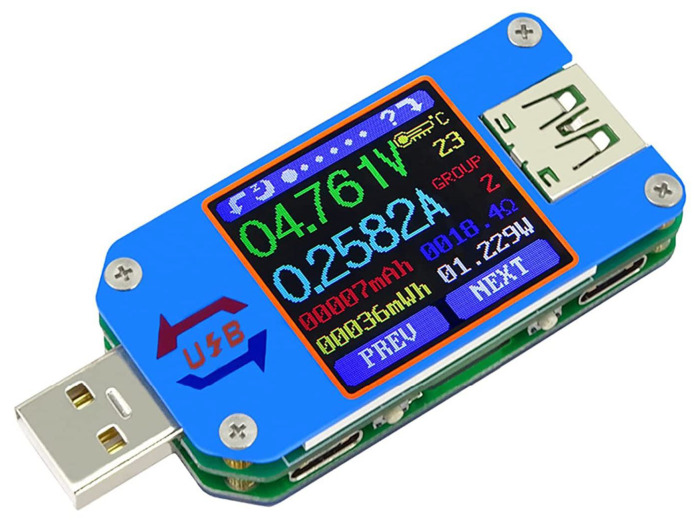
USB Power Meter Tester used to estimate power consumption.

**Figure 5 sensors-23-01039-f005:**
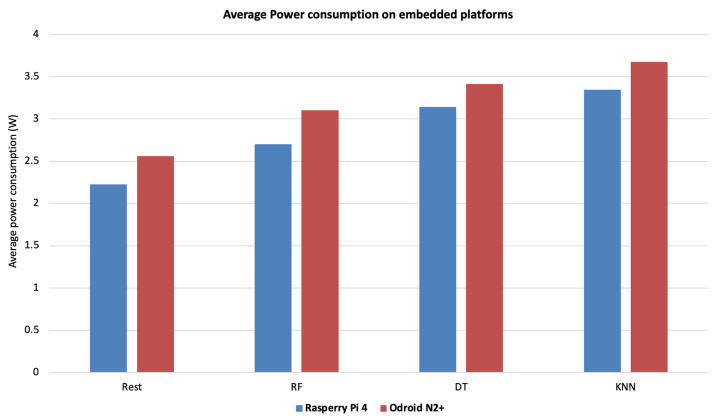
Average power consumption (W) of rest phase and of each classifiers on the embedded boards.

**Figure 6 sensors-23-01039-f006:**
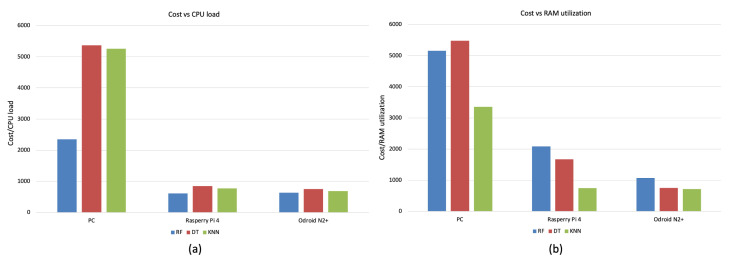
Average power consumption (W) of rest phase and of each classifiers on the embedded boards.

**Table 1 sensors-23-01039-t001:** Main Features of Shimmer3 IMU.

Features	
Dimensions	51 mm × 34 mm × 14 mm
Weight	23.6 g
Microcontroller	24 MHz TI MSP 430
Tri-axial accelerometer	Kionix KXTC9-2050
Acceleration range	±2 g
Acceleration sensitivity	660 mV/g (±20 mV)
Wireless connectivity	Bluetooth (IEEE 802.15.1)
Local storage	8 GB microSD card
Battery	Li-ion battery
Sampling rate	selectable up to 1024 Hz

**Table 2 sensors-23-01039-t002:** Main Features of Shimmer3 IMU.

Features	
Mean Absolute Value	$\frac{\sum_{i=1}^{N}\left|Acc_{i}\right|}{N}$
Standard Deviation (σ)	$\sqrt{\frac{\sum_{i=1}^{N}{\left(Acc_{i}-\mu \right)}^{2}}{N}}$ Where μ is mean of considered temporal window
Variance (VAR)	$\frac{\sum_{i=1}^{N}{\left(Acc_{i}-\mu \right)}^{2}}{N}$ Where μ is mean of considered temporal window
Maximum	$\max(Acc_{i})$
Minimum	$\min(Acc_{i})$
Root Mean Square	$\sqrt{\frac{\sum_{i=1}^{N}Acc_{i}^{2}}{N}}$
Simple Squared Integral	$\sum_{i=1}^{N}Acc_{i}^{2}$
Wavelet Entropy	$-\sum_{i=1}^{N}Acc_{i}*\ln\left(Acc_{i}\right)$
Skewness	$\frac{\sum_{i=1}^{N}{\left(Acc_{i}-\mu \right)}^{3}}{N*\sigma }$ Where μ is mean and σ is the VAR of considered temporal window
Kurtosis	$\frac{\sum_{i=1}^{N}{\left(Acc_{i}-\mu \right)}^{4}}{N*\sigma }$ Where μ is mean and σ is the VAR of considered temporal window
Dynamic Acceleration Change	$\max\left(Acc_{i}\right)-\min\left(Acc_{i}\right)$
Static Acceleration Change	$\max\left(FilteredAcc_{i}\right)-\min\left(FilteredAcc_{i}\right)$
Log Energy Entropy	$\sum_{i=1}^{N}\log_{2}\left(Acc_{i}\right)$

**Table 3 sensors-23-01039-t003:** Optimal parameters selected for each classification models.

Model	Parameters
RF	max_depth = 30, n_estimators = 25, criterion = gini
DT	criterion = gini, max_depth=19
KNN	n_neighbors = 13, metric = minkowski, algorithms = auto, weights = distance

**Table 4 sensors-23-01039-t004:** Comparison of embedded systems versus PC.

Hardware	PC	Raspberry	Odroid
Model	Lenovo ThinkCentre M70s Tiny	Pi 4 Model B	N2+
CPU	Intel Core i5	Quad Core ARM Cortex-A72	Quad Core ARM Cortex-A73
RAM	8 Gb	8 Gb	4 Gb
Connectivity	Bluetooth, Wifi, Ethernet, USB	Bluetooth, Wifi, Ethernet, USB	Bluetooth with adapter, Wifi, Ethernet, USB
Video output	HDMI	mini HDMI	HDMI
Storage	256 GB SSD	32 GB SD-Card	32 GB SD-Card
Dimensions	340 × 298 × 92.5 mm	88 × 58 × 19.5 mm	90 × 90 × 17 mm
Weight	5200 g	46 g	200 g
Energy consumption	180–200 W	2–6 W	2.2–6.2 W
Operating Voltage	AC 220 V/DC 19 V	AC 220 V/DC 5 V	AC 220 V/DC 12 V
Operating system	Windows 10	Raspbian	Ubuntu
Cost	789	200	199

**Table 5 sensors-23-01039-t005:** Classifier results with considered metrics.

Model	Acc	Pr	Re	F1
RF	0.987	0.986	0.986	0.977
DT	0.971	0.973	0.971	0.972
KNN	0.943	0.942	0.943	0.942

**Table 6 sensors-23-01039-t006:** Average time (sec) to obtain a posture varying classifiers and platforms.

Model	PC	Raspberry Pi 4	Odroid N2+
RF	0.017	0.044	0.055
DT	0.002	0.005	0.008
KNN	0.004	0.012	0.015

**Table 7 sensors-23-01039-t007:** Classifier results for RF with downsampling to 25 Hz.

Model	Accuracy	Precision	Recall	F1
RF	0.962	0.943	0.941	0.942

**Table 8 sensors-23-01039-t008:** CPU Load (%) and Memory utilization (%) of every classification models across all considered hardware platforms.

Model	PC		Raspberry Pi 4		Odroid N2+	
	CPU	RAM	CPU	RAM	CPU	RAM
RF	0.332	0.153	0.324	0.096	0.312	0.186
DT	0.147	0.144	0.237	0.120	0.265	0.264
KNN	0.150	0.235	0.258	0.268	0.289	0.279

## Data Availability

The data presented in this study are available on request from the corresponding author. The data are not publicly available due to restrictions (their containing information that could compromise the privacy of research participants).

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
