# Peer review of "Human Postures Recognition by Accelerometer Sensor and ML Architecture Integrated in Embedded Platforms: Benchmarking and Performance Evaluation"

_sensors, 2023, doi:10.3390/s23021039_

Round 1
Reviewer 1 Report
Very interesting paper that can ameliorate the knowledge in this sector
Author Response
Dear reviewer,
thank you for the kind evaluation! The paper has been improved with an english style refining.
Reviewer 2 Report
Congratulations to the authors, the research Human Postures Recognition by Accelerometer Sensor and ML Architecture Integrated in Embedded Platforms: Benchmarking and Performance Evaluation can improve analysis in sports.
Author Response
Dear reviewer,
thank you for the kind evaluation!
Reviewer 3 Report
Dear authors,
The manuscript is written well, so my decision is acceptance.
Author Response

(The authors gave the same response as above.)
